# Injuries and Overuse Injuries in Show Jumping—A Retrospective Epidemiological Cross-Sectional Study of Show Jumpers in Germany

**DOI:** 10.3390/ijerph19042305

**Published:** 2022-02-17

**Authors:** Heinz-Lothar Meyer, Philip Scheidgen, Christina Polan, Paula Beck, Bastian Mester, Max Daniel Kauther, Marcel Dudda, Manuel Burggraf

**Affiliations:** 1Department of Trauma, Hand and Reconstructive Surgery, University Hospital Essen, 45147 Essen, Germany; philip-scheidgen@web.de (P.S.); christina.polan@uk-essen.de (C.P.); paula.beck@uk-essen.de (P.B.); bastian.mester@uk-essen.de (B.M.); marcel.dudda@uk-essen.de (M.D.); manuel.burggraf@uk-essen.de (M.B.); 2Department of Trauma Surgery, Orthopedics and Pediatric Orthopedics, Agaplesion Diakonieklinikum Rotenburg, 27356 Rotenburg, Germany; uch@diako-online.de

**Keywords:** epidemiology, equestrian sports, trauma, sports injuries, show jumping

## Abstract

This retrospective cross-sectional epidemiological study deals with sport-specific injury patterns in show jumping. A total of 363 show jumpers of all levels (S) answered a retrospective questionnaire about injuries and overuse damages which occurred in the course of their careers. Demographic data and information on injuries in various body regions were collected. In addition to descriptive analysis, significance tests were performed. For better statistical comparability with other sports, exposure time was extrapolated with total career duration and weekly training hours, and injuries per 1000 jumping hours were calculated. The study included 251 (69%) women and 112 (31%) men, who were on average 26.9 ± 10.9 years old. The injury rate for the entire collective was 3.7 per 1000 h of exposure. The most frequently affected body region was the head (31%). Overuse complaints play a subordinate role and mainly affect the upper extremities (65%). The riders of the professional lower performance levels are less likely to injure themselves per 1000 h than riders of the higher performance levels. Riders who often or always wore a helmet suffered significantly fewer head injuries (*p* = 0.008) and had a significantly lower total injury duration than riders who did not wear a helmet (*p* = 0.006). Similarly, the study showed that riders who often or always wore a safety vest suffered significantly fewer spinal injuries (*p* = 0.017) and had significantly fewer injuries per 1000 riding hours (*p* = 0.031) than riders who did not wear a safety vest. Based on the present results, there should be an extension of the general helmet requirement and a requirement to wear safety vests in show jumping in general.

## 1. Introduction

Show jumping is a discipline of equestrian sport in which various obstacles in a marked area as a course must be overcome in a specified sequence and in a specified time by horse and rider [1,2]. The height, type and distance of the various obstacles have a significant influence on the level of difficulty. An obstacle can be set up as a single obstacle or as a combination of different obstacles. Additionally, the difficulty level is influenced by the distances of the obstacles. A good equestrian training and a high quality and fitting material equipment is crucial. As “proper equipment for horse and rider” the German Equestrian Federation gives the following basic equipment: Riding helmet, riding breeches, riding boots, riding gloves, protective vest, saddle (with girth, stirrups, stirrup leathers, stirrup lock and saddle pad), snaffle (with bit and halter), auxiliary reins and leg protection [1,2]. Both sexes compete in the same category [2]. The performance level (S) in show jumping divides the riders into different groups according to successes, special qualifications (e.g., horse manager, trainer qualifications) and riding badges taken. The classification ranks from S7 (trial license) to the best performance level S1 in descending order. In principle, all performance classes are permitted in competitions; however, the text of the announcement of the respective competition is decisive [2,3].

Equestrian sport in general is associated with multiple possibilities for injury and the competitive sport in particular can be classified as a risky sport. Show jumping ranks as one of the most accident-prone disciplines [4,5]. Serious polytrauma results predominantly from combination falls of horse and rider. An inversely proportional correlation between equestrian experience and the frequency of equestrian accidents is described [5].

Since January 2021, a general helmet obligation has been introduced for almost all equestrian disciplines [2]. According to data, only about 9–40% of riders wear a helmet during an accident, although the risk of head and neck injuries is reduced fivefold [6,7].

Research on current publications related to horseback riding revealed some general research on horseback riding injuries of all riding disciplines. To date, there is hardly any literature considering injuries specific to show jumping and different riding levels in show jumping [8,9]. Lechler et al. observed an injury rate between 1.1 and 2.1 injuries per 1000 h of riding [8].

The aim of this study is to record and evaluate sport-specific overuse and injury patterns of all body regions and injured structures in show jumpers of all performance levels, as well as differences between genders and protective equipment. The collected data should lead to a further development of preventive as well as protective measures to make the sport safer.

## 2. Materials and Methods

The retrospective cross-sectional epidemiological study was carried out using a questionnaire analogous to previous studies in accordance with the Helsinki Declaration and after review by the responsible ethics committee of the University of Duisburg-Essen (09-4123-BO) [10,11,12]. The questionnaires were distributed to 363 active show jumpers of different performance levels and filled in straightaway under the supervision of medical staff who were available to answer questions and explain ambiguities. The questionnaires were filled out voluntarily and anonymously. Every rider in possession of a valid FN annual competition license (riding license) with the performance level endorsement S6 to S1 could participate in the survey. Trial license holders (S7) were excluded for better comparability. Participants were randomly selected at official show jumping competitions in Germany. The age range of the participants was between 10 and 69 years. Demographic data and information about injuries in eight different body regions were collected. In addition, for better statistical comparability with other sports, exposure time was extrapolated with total career duration and weekly training hours, and injuries per 1000 jumping hours were calculated.

### Statistical Analysis

The statistical evaluation was carried out using IBM SPSS Statistics 25 software (IBM, Armonk, NY, USA). Within the context of descriptive statistics, a calculation of the mean value and the standard deviation (M ± SD) was performed. All values were tested for normal distribution with the Kolmogorov–Smirnov test. In addition to the descriptive evaluation of the number of injuries in the different body areas, the influence of the variables, gender and performance level, on the number of injuries per body area was also investigated. For this purpose, the sum of the injuries was first calculated for each body area from the number of injuries relevant to the body area in each case. Subsequently, significance tests were performed with this sum variable. Due to the non-normal distribution of the sum variable, non-parametric tests were used. For the dichotomous variable gender, a Mann–Whitney *U* test was used in each case. For the multilevel variables riding experience and performance class, Kruskal–Wallis tests were performed first in each case. If this revealed a significant difference between at least two groups, pairwise Mann–Whitney U tests were then calculated to investigate between which groups there were significant differences in the number of injuries. Values of *p* < 0.05 were considered as significant.

## 3. Results

A total of 363 riders participated in the study, divided into 251 (69.1%) female and 112 (30.9%) male riders. The mean age of all participating athletes was 26.9 ± 10.9 years, that of the female participants 25 ± 8.5 years and that of the male participants 31.3 ± 14.2 years. The mean body weight was 66.4 ± 3.2 kg and the mean body mass index was 22.1 ± 3.2 kg/m^2^. The respondents practiced show jumping for a mean of 12.7 ± 9.9 years. In total, 1,832,558 riding hours (=exposure time) were reported. The average exposure time per rider was 5048 h. On all indicated variables an injury rate of 3.7 per 1000 riding hours can be found. The distribution of the study participants according to performance level is shown in Table 1. The average reported training time is 5.5 ± 9.1 h per week. Study participants reported a total of 15,215 falls. The average rider fell 41.9 ± 110.9 times. The majority of these falls occurred during training with 12,788 (84.1%) falls. On average, this resulted in 35.2 ± 104.6 falls and an average of 3.4 ± 6.7 falls per horse jumping year. Per training year, each athlete fell on average 2.8 ± 6.2 times and on average 0.6 ± 1.1 times at a competition. For every 1000 riding hours, 8.3 falls were reported in the study group. In the range of performance level, there are significant differences of riding hours (*p* < 0.001). From S 5 to S 2, the number of riding hours increases significantly. S 1 rides significantly less than S 2 (Table 1). The injury-related training breaks of all participants in total amount to 2418.1 weeks. On average, each athlete had to take 6.7 ± 17.0 weeks of injury-related breaks in their career to date and had to take an injury-related break of 0.5 ± 1.2 weeks per career year. Spinal injuries led to the longest injury-related time off and account for 27.3% of total downtime.

### 3.1. Injury Location

The injury location and frequency showed that 326 (89.8%) athletes had been injured at least once since they started practicing the sport. A total of 6768 injuries and overuse injuries were reported in the entire collective. The most common location was the head, followed by the trunk. Variables classified as overuse accounted for only 4.4% (296) of all mentions. In the following, all injuries and overloads are considered by body region with significance testing of the number of injuries per performance level. The consideration of the mentioned variables regarding the total riding duration was carried out in advance. With 11,648.5 ± 19,069.3 h on average, men rode significantly more hours than women with 2103.3 ± 5125.4 h (*p* < 0.001).

#### 3.1.1. Head

A total of 2073 head injuries were reported and 232 (63.9%) riders sustained at least one head injury during their career. Based on the complete study collective, this results in an average of 5.7 ± 11.3 head injuries. Figure 1 shows all recorded head injuries. Men have significantly more head injuries than women in their riding careers (*p* < 0.05). There are also significant differences in the number of head injuries ever in the performance level (*p* < 0.001). Significantly more injuries are shown in S 1 than in S 3 to 6, significantly more in S 2 than in S 5 and 6 and significantly more in each of S 3 and 4 than in S 6. Considering 1000 h of exposure, no significant differences can be shown (*p* > 0.05). In summary, riders in better performance levels have more head injuries (Figure 2).

#### 3.1.2. Trunk

A total of 1612 truncal injuries were reported and 247 (68%) athletes injured their trunk at least once in their career. On average, every athlete sustains a truncal injury 4.4 ± 7.6 times in his career. Figure 3 shows all recorded truncal injuries. Men suffer significantly more trunk injuries in their career than women (*p* < 0.05). For every 1000 h of exposure, women are significantly more likely to sustain truncal injuries (*p* < 0.05). Divided into performance levels, there are significant differences regarding the number of truncal injuries over the course of the entire career (*p* < 0.001). There were no significant differences between S 1 to S 3 over the entire riding career and significantly fewer injuries in each case compared to S 4–6. The S 4–6 do not differ significantly from each other. In summary, riders in the lower performance levels sustain more truncal injuries than riders in the higher performance levels (Figure 4).

The number of injuries to the remaining body regions are shown in Table 2.

### 3.2. Types of Injuries

Contusions and skin injuries were the most common types of injuries in show jumping, followed by distensions and distortions. All injury types are shown in Table 3. Overuse and inflammation occurred in 20.4% (*n* = 74) of the samples. More than half were tendinitis of the elbow (51.3%), followed by carpal tunnel syndrome (14.3%). Comparing genders, there were no significant differences in neither complete career (*p* > 0.05) nor 1000 h of exposure (*p* > 0.05). The same applies to the performance levels ever (*p* > 0.05). The age categories differ significantly in the number of overloads in the entire career (*p* < 0.05), especially for more overloads in the over 51-year-olds compared to all under 40-year-olds. For the whole career, there were significant differences in riding experience (*p* < 0.05). Riders with a career of more than 20 years had significantly more overloads than riders with less than 10 years of riding experience.

### 3.3. Injury Break

The injury-related training breaks for all participants in total amount to 2418.1 weeks. On average, each athlete had to take a break of 6.7 ± 17.0 weeks in their career due to injuries. The longest downtime was 214 weeks. On average, each rider had to take an injury-related break of 0.5 ± 1.2 weeks per career year. Spinal injuries accounted for the longest downtime, at 27.3% of total downtime. Lower leg and foot injuries at 15.6% and knee injuries at 14.5% combine for about another third of all injury time off. In descending order, shoulder injuries (11.4%), pelvic and thigh injuries (10%) and head injuries (9.2%) are similarly likely to trigger riding breaks. Hand (7.4%) and elbow injuries (4.5%) accounted for the lowest percentage of all injury-related breaks.

### 3.4. Protective Equipment

Of the 363 jumpers surveyed, 86 (24%) reported to never or occasionally wear a helmet and 277 (76%) riders reported to often or always wear a helmet. In the present study riders with helmets suffered 4.7 ± 10.1 head injuries and riders without helmets suffered 9.1 ± 14.1 head injuries. This indicates that riders who often or always wear a helmet suffer significantly fewer head injuries than riders who do not or only occasionally wear a helmet (*p* = 0.008). Similarly, the present study shows that riders with helmets reported a significantly (*p* = 0.006) lower total injury duration (15.1 ± 28.5 days) than riders without helmets (26.2 ± 32.8 days). No significant difference was seen in the number of days of training lost due to head injuries for riders with or without helmets (*p* = 0.318). Regarding to the total number of injuries per 1000 h of riding, no significant difference was shown between the groups of riders with and without helmets (*p* = 0.744).

Always or often wearing a safety vest was reported by 50 (14%) riders and 313 (86%) never or occasionally wear a safety vest. Riders with safety vests suffered a mean of 2.9 ± 4.3 and riders without safety vests suffered 4.7 ± 7.9 spinal injuries. Thus, riders who often or always wore a safety vest suffered significantly fewer spinal injuries than riders who occasionally or never wore a safety vest (*p* = 0.017). In the total number of injuries per 1000 riding hours, riders without a safety vest also had significantly more injuries than riders with a safety vest (*p* = 0.031).

## 4. Discussion

Show jumping has become a popular sport. However, there is hardly any explicit sports science data on injuries in show jumping in the literature, to date. Lechler et al. describe a total of 636 injuries in show jumping in a cross-sectional survey of 264 show jumpers [8]. The show jumping collective of Lechler et al. uncharacteristically consists of 50.4% male riders [8]. In the present study, male participants account for 30.9% of the participants which is more prevalent than other discipline-independent equestrian studies at 12 to 13% [13,14]. In publications with larger athlete collectives, the average age of the athletes is 25.7 to 26.6 years, very similar to the present study [13,14]. The show jumping collective of Lechler et al. is also comparable with an average of 24.7 years [8]. The average riding career in the present study of 12.7 years is shorter than in Lechler et al. but slightly longer than in other general riding studies of 9.4 to 12 years [8,15]. The riding training time of the total collective of 5.5 h per week is similar to the 5.7 to 6 h in the general comparison studies but significantly lower than Lechler et al., who reported 10.6 h per week for amateurs and 31.2 h per week for professional show jumpers [8,15]. In the present work, a classification according to performance levels was made with the aim of a discriminatory power between inexperienced and experienced riders. In the comparative literature, there is often a distinction between amateur and professional or no survey of the riding levels [8].

### 4.1. Injury Frequency

For the most part, the injury rate in the present study is higher than that found elsewhere in equestrian sports, at 3.7 per 1000 h of riding. Lechler et al. give an injury rate in show jumping of 1.1 per 1000 h for professional riders and 2.1 per 1000 h for amateur riders but with only 636 total injuries [8]. The lower injury rate in show jumping by more professional riders was confirmed in the present study. A large retrospective study by Mayberry et al. found that novice riders with less than three years of experience have a five times higher risk of injury than advanced riders and eight times higher than professional riders. A sharp decrease in injury frequency has been observed after only 18–100 h of riding experience [16]. This suggests that riding experience has a preventive influence on suffering a riding injury. This was confirmed in the present study.

However, while novice riders have been shown to have a higher overall risk of injury, professional riders suffer more severe injuries. This is likely a result of the higher level of difficulty at which they train and compete and are pushed to significantly higher speeds and higher obstacles by competing in competition and also in training. There is a tendency, especially among professionals, to ride and train less experienced or more volatile horses [16,17]. Other comparative literature suggests that this can be explained by the higher number of hours of loading in this group, which was confirmed in the present study [8,18]. Performance level 1 and 2 riders ride approximately four times more hours than performance level 3 riders and than the overall average. Performance level 6 riders ride on average only 2.3% of the riding time compared to performance level 1. Professional show jumpers presumably have more experience and can ride more safely [5]. In almost all categories this results in the more professional riders having less injuries per 1000 exposure hours. The present study confirms that professionalization leads to fewer injuries in terms of exposure time. Injury severity and horse experience were not considered decisively in the present study.

In equestrian sports, the incidence of injury is thought to be approximately 0.5 injuries per 1000 h [19]. In race riding, Mc Crory et al. describe a fall incidence of 0.1% per ride with 41.4% resulting injuries from the fall [20]. However, compared to other sports, the injury rate is lower than, for example, contact sports. Injuries in equestrian sports must be considered as particularly severe injuries compared to other sports and are comparable to injuries in skating or cycling [21].

### 4.2. Injury Location

A total of 6768 injuries and overloads have been recorded in the total collective. Of these, 5% were overloads. The most common injury location was the head (30.6%), followed by injuries to the trunk (23.8%). The most commonly reported injury types are contusions and skin injuries (47.4%). Minor injuries such as contusions and skin injuries are the leading injury type in general equestrian sports and account for 44% to 51% of injuries [14,22]. In the present study, 63.9% of all riders suffer head injuries and thus account for 30.6% of all injuries. Different values on head injuries in equestrian sports are reported in the literature. Lechler et al. describe about 20% of injuries in the area of the head in show jumping [8].

In the present study the trunk injuries were the second most frequently injured region with 24% and lead to the longest injury break time. Compared to other studies, this injury is underrepresented in our study [23,24,25].

According to the literature, most injuries involve the upper extremity in equestrian sports [5,13,26]. In summary, in this study, as in Lechler et al., upper extremity injuries account for 26.9% of all injuries but are not leading [8]. The cause of many injuries in this region is the attempt to catch oneself in a fall with outstretched arms or the handling of the reins while riding [22,27].

### 4.3. Differences of the Collective

In the present study, significantly more men than women were injured in the head region (*p* < 0.05). On the one hand, this could be related to the fact that men are presumably more risk averse than women and that men are less likely to wear a helmet than women [28,29]. This contradicts the findings of Havliks et al., who observed that women were an independent risk factor for equestrian-related injuries. However, this could be biased by the predominance of female riders, especially in recreational and amateur riding [17]. In terms of hours of exposure, women are probably more likely to be injured, but this cannot be demonstrated in this study. In addition, a study by Krüger et al. in which retrospective data were collected from all equine-related accidents at a German Level I Trauma Centre took place between 2004 and 2014. Of the 770 injured patients, 87.9% were female [13]. Here, however, no distinction was made according to the injured body region.

The differentiated consideration of injuries per 1000 riding hours of the individual performance levels shows fewer injuries in performance levels 1 and 2. This is comparable to the known higher frequency of injuries in amateur riders in comparable studies. Here, professional jumpers had more severe injuries than amateur athletes (see Section 4.1 injury frequency) [5,8].

### 4.4. Types of Injuries

Literature data for fractures in general equestrian sports vary widely from 3 to 40% [14,22,30]. In the present study, only 3.6% fractures were recorded. The discrepancies possibly result from the absence of very young riders with many fractures [22]. Comparison of dislocations as well as distensions and distortions with data from discipline-independent studies yields similar frequencies [15]. Overuse injuries play a minor role compared to injuries with 335 indications in only 20% of the riders. More than half of the data refer to inflammations of the tendon sheaths at the elbow, which are mainly caused by repetitive strain on the finger extensors [31].

### 4.5. Protective Equipment

The risk of injury can be relevantly reduced by wearing protectors in sports [6,7]. In equestrian sports, this mainly concerns wearing riding helmets [6]. The literature shows that 9 to 23% of riders in general equestrian sports wear a helmet [7,32]. In the present study, 76% (277) of the riders reported to often or always wear a helmet. However, in the literature, the number of injuries to the head remains unaffected by this [4]. We were able to disprove this in the present work. Riders who often or always wore a helmet suffered significantly fewer head injuries than riders who did not or only occasionally wore a helmet (*p* = 0.008). Similarly, riders with helmets had a significantly lower total injury duration than riders without helmets (*p* = 0.006). Regarding the total number of injuries per 1000 h of riding, no significant difference was shown between the groups of riders with and without helmets (*p* = 0.744). This is related to the fact that the protective equipment only protects one body region at a time. In 2010, a general helmet requirement was introduced in almost all equestrian sports [2].

Another form of protection is the safety vest. In this study, 86% (313) of the riders reported to never or occasionally wear a safety vest. These figures are consistent with those in the literature [4,25]. Reasons given for infrequent use included limited range of motion, sweating and aesthetic reasons [25]. Evidence that protective vests have a protective function has not been provided to date, and in the study by Hessler et al., riders wearing a vest actually sustained more injuries to the thorax, abdomen and spine [4]. This could be disproved in the present study. Riders who often or always wore a safety vest sustained significantly fewer spinal injuries than riders who occasionally or never wore a safety vest (*p* = 0.017). Moreover, in the total number of injuries per 1000 riding hours, riders without a safety vest had significantly more injuries than riders with a safety vest (*p* = 0.031). A general recommendation to wear a vest has not yet been made. Authors should discuss the results and how they can be interpreted from the perspective of previous studies and of the working hypotheses. The findings and their implications should be discussed in the broadest context possible. Future research directions may also be highlighted.

## 5. Conclusions

In the present study, we were able to obtain a good overview of injuries and overuse injuries in show jumping. The injury rate for show jumping was higher than in comparable studies on equestrian sports but lower than in other sports such as contact sports, with 3.7 injuries per 1000 h of riding. [19,21]. The most common injury location was the head (30.6%), followed by injuries to the trunk (23.8%). Overuse complaints play a subordinate role and mainly affect the upper extremities (65%). The riders of the professional lower performance levels are less likely to injure themselves per 1000 h than riders of the higher performance levels.

This study was able to show that riders who often or always wore a helmet suffered significantly fewer head injuries (*p* = 0.008) and had a significantly lower total injury duration than riders who did not wear a helmet (*p* = 0.006). Similarly, the study showed that riders who often or always wore a safety vest suffered significantly fewer spinal injuries (*p* = 0.017) and had significantly fewer injuries per 1000 riding hours (*p* = 0.031) than riders who did not wear a safety vest. In general, these results confirm the introduction of mandatory helmet use in show jumping competitions. However, based on the present results, there should be an extension of the general helmet requirement and a requirement to wear safety vests in show jumping in general. According to the present study, this could prevent many injuries in show jumping.

## Figures and Tables

**Figure 1 ijerph-19-02305-f001:**
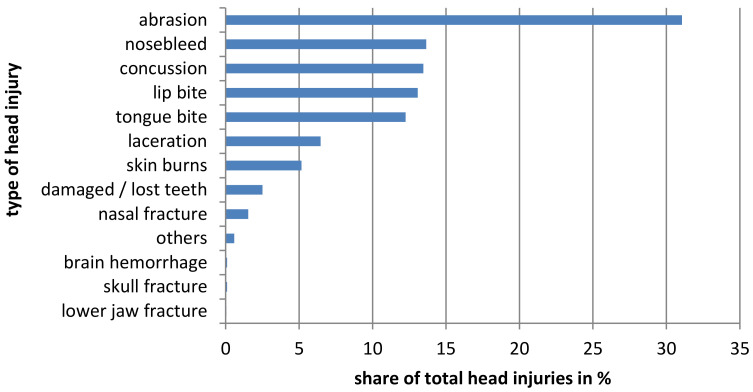
All recorded head injuries are shown.

**Figure 2 ijerph-19-02305-f002:**
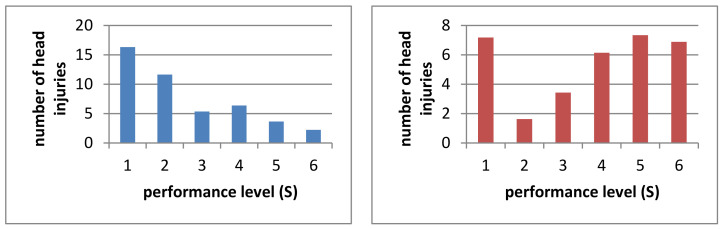
Number of head injuries by performance level ever (blue) and in 1000 h (red).

**Figure 3 ijerph-19-02305-f003:**
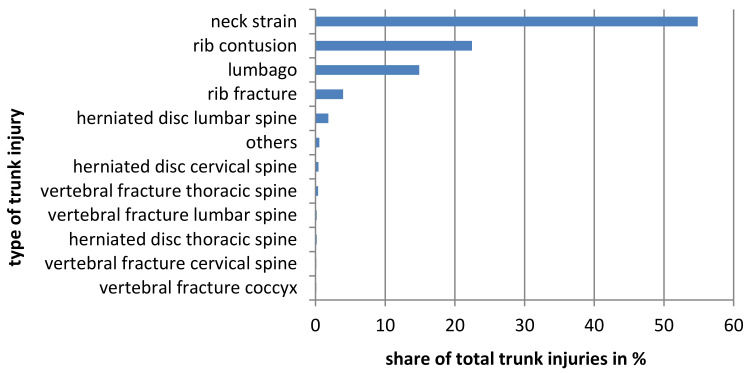
All recorded trunk injuries are shown.

**Figure 4 ijerph-19-02305-f004:**
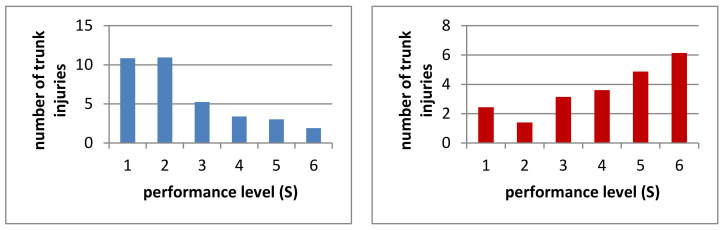
Number of trunk injuries by performance level ever (blue) and on 1000 h (red).

**Table 1 ijerph-19-02305-t001:** The distribution of the study participants according to performance level.

Performance Level	Number of Study Participants	Age in Years	Riding Hours (Exposure Time)	Injuries/1000 Riding Hours
S 1	18 (5.0%)	29.9 ± 9.6	20,716.2 ± 23,546	2.1
S 2	33 (9.1%)	34.3 ± 14.8	22,279.6 ± 25,901.2	1.7
S 3	69 (19.9%)	30.9 ± 13.5	5556.5 ± 8289.3	3.6
S 4	85 (23.4%)	25.5 ± 8.2	1850.3 ± 1755.8	11.1
S 5	92 (25.3%)	26.2 ± 9.6	1657.8 ± 2586.8	6.9
S 6	66 (18.2%)	21.1 ± 6.7	473.5 ± 407.1	17.6

**Table 2 ijerph-19-02305-t002:** All recorded injuries to the remaining body regions are shown.

Body Region	Total Number of Injuries/Overuse Damage	Number of Riders with Injuries	Average Number of Injuries/Careers Per Rider
head	2073 (30.6%)	232 (63.9%)	5.7 ± 11.3
trunk	1612 (23.9%)	247 (68%)	4.4 ± 7.6
shoulder	607 (9.0%)	150 (41.3%)	1.7 ± 4.3
elbow	329 (4.9%)	88 (24.2%)	0.9 ± 3
wrist/hand	883 (13.0%)	162 (44.6%)	2.4 ± 7
pelvis/thigh	328 (4.8%)	101 (27.8%)	0.9 ± 2.4
knee	595 (8.8%)	122 (33.6%)	1.6 ± 10.8
lower leg	341 (5.0%)	99 (27.3%)	0.9 ± 2.7

**Table 3 ijerph-19-02305-t003:** Distribution of injury types in show jumping.

Injury Type	Number of Injuries	Average Number of Injuries/Career of a Rider	Most Common Injury
bruises/skin injuries	3209	8.8 ± 16.4	head abrasion (20.1%)
distensions/distortions	1885	5.2 ± 9	cervical spine distortion (46.9%)
dislocations	275	0.8 ± 10.5	patellar dislocation (82.2%)
fractures	246	0.7 ± 1.6	rib fractures (26%)
ruptures	196	0.5 ± 1.9	hand tendon injuries (42.4%)

## Data Availability

The data presented in this study are available on request from the corresponding author.

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
