# Peer review of "Injuries and Overuse Injuries in Show Jumping—A Retrospective Epidemiological Cross-Sectional Study of Show Jumpers in Germany"

_ijerph, 2022, doi:10.3390/ijerph19042305_

Round 1

Reviewer 1 Report

General Comments

This study’s objective was to collect and evaluate sport-specific overuse and injury patterns in show jumping riders of all performance levels. It was determined that overuse injuries mainly affect the upper extremities and were minimal in occurrence compared to acute injuries. The most frequently injured body part was the head followed by trunk injuries. This study also found higher level riders have more injuries in total but fewer injuries per 1000 riding hours compared to lower-level riders. Additionally, this study also demonstrated that the use of helmets and safety vests significantly reduced the number of injuries sustained by riders. Given these results it was recommended that there should be an extension of the general helmet requirement and wearing a safety vest should also be mandatory in show jumping.

This study can provide useful information to current literature but in the manuscript’s current form there is lack of focus and interpretation of the studies results. The author’s primary objective and title of the manuscript suggest the study is focused on sport-specific overuse and injury patterns in show jumping riders of all performance level, however little interpretation of these results are provided. Given this is the primary objective of the study a more in-depth analysis and discussion of rider performance levels and their relation to overuse and acute injury patterns is required. The authors also present many other results such as the effect of gender but do not provide any meaningful discussion of these results. It is important that the authors discuss and interrupt all comparisons presented in the results section.

The main findings and recommendations highlighted by the authors in the current manuscript are related to the effect of protective equipment (i.e. helmets and safety vests) on injury rates. While a larger discussion is provided surrounding the effect of protective equipment on injury rates, further analysis and discussion of these results could improve the study.

It is recommended that the authors provide more discussion surrounding the abundance of results presented in the current study. Alternatively, the study could be revised to focus on the effect of protective equipment on injury rates in show jumping as these results appear to be the main findings and recommendations highlighted by the authors.

Specific Comments

Abstract

  1. p. 1. “The riders of the professional lower performance levels are less likely to injure themselves per 1000 hours than riders of the higher performance levels.”
    1. This sentence is not supported by the data of this study and the subsequent text in the other sections of this manuscript. Please revise accordingly as the data and rest of the manuscript say the opposite. Additionally, explain why this finding is important.

Introduction

  1. p. 1. “Equestrian sport in general is associated with multiple possibilities for injury and especially the competitive sport must be classified as a risky sport.”
    1. Rephrase to “… and the competitive sport in particular can be classified as a risky sport.”
  2. p. 1 “Show jumping ranks as the most accident-prone discipline after trail riding and dressage [3, 4].”
    1. Are the authors stating that trail riding and dressage are the most accident-prone discipline in equestrian sports? Based on research I have reviewed horse racing and cross-country would be more accident-prone than trail riding and dressage. To my knowledge trail riding and dressage would be among the safest of equestrian disciplines. Unfortunately, I do not have access to the cited papers and cannot read German so I cannot comment any further.

Material and methods

            Statistical analysis

  1. p. 2. “The two-sided t-test was used for normal distributed values. For non normally distributed values, the Mann-Whitney-U test for non-parametric data was used to detect differences between unconnected test groups.”
    1. Where t-tests and Mann-Whitney-U tests used to compare differences between performance levels?
    2. If so, the authors should use ANVOAs and Kruskal-Wallis or Mood tests and subsequent post hoc tests when comparing performance levels instead of -tests and Mann-Whitney-U tests?

Results

  1. p. 2. “From LK 5 to LK 2, the number of riding hours increases significantly. LK 1 rides significantly less than LK 2 (Table 1).”
    1. Should LK 5, LK 2 and LK 1 be S5, S2 and S1 for consistency with previously stated classification of performance levels?
    2. Similar should Table 1 read S1 – S6 instead of LK 1 – LK 6?
    3. Please revise text in the manuscript to ensure nomenclature is consistent throughout the manuscript.

Protective equipment

  1. p. 6. “In the present study riders with helmets suffered 4.7 ± 10.1 and riders without helmets suffered 9.1 ± 14.1 head injuries.”
    1. Add “head injuries” after 4.7 ± 10.1.

Discussion

  1. p. 6 – 7. “In the present study, male participants are more prevalent at 30.9% than in discipline-independent equestrian studies at 12 to 13% [10, 11]. The show jumping collective of Lechler et al uncharacteristically consists of 50.4% male riders [6].”
    1. Change the order of these sentences to allow the text of the manuscript to read better.
    2. Additionally revise the first sentence accordingly “In the present study, male participants account for 30.9% of the participants which is more prevalent than other discipline-independent equestrian studies at 12 to 13% [10, 11].”

Injury frequency

  1. p. 7. “The lower injury rate in show jumping by more professional riders was confirmed in the present study.”
    1. Can the authors use the data in their study and the literature to discuss why the injury rate is lower for higher level riders? Given the context of this studies analysis this discuss seem more important than comparing show jumping to other equestrian disciplines and other sports.

Differences of the collective

  1. p. 7. “In the present study, significantly more men than women were injured in the head region (p < 0.05). This contradicts the findings of Havliks et al who showed that women are an independent risk factor in horseback riding only by numerical superiority [13]. In relation to exposure hours, women are probably more frequently injured, but without evidence of significance in this study.”
    1. This paragraph and the corresponding data require further analysis and discussion. At present the authors have not provided any meaningful analysis or interpretation of the current studies results in context of the literature

Protective equipment

  1. p. 8. Was the head injury rate per 1000 hours of riding different between the groups of riders with and without helmets? These results would be more interesting and meaningful to discuss rather than total number of injuries or total number of injuries per 1000 hours of riding.

  1. p. 8. “Authors should discuss the results and how they can be interpreted from the perspective of previous studies and of the working hypotheses. The findings and their implications should be discussed in the broadest context possible.”
    1. Yes, please take the opportunity to have this discussion in the current paper. The current study already has the data and results to discuss this topic.

Reviewer 2 Report

Injuries and overuse damage in show jumping - a retrospective

cross-sectional epidemiological study

This retrospective cross-sectional epidemiological study examined sport-specific injury patterns in show jumping. A total of 363 show jumpers of all levels answered a retrospective questionnaire about injuries and overuse damages which occurred in the course of their careers. The injury rate for the entire collective was 3.7 per 1000 hours of exposure. The most frequently affected body region was the head (31%). Overuse complaints mainly affected the upper extremities (65%). Serious injuries were. However, riders of lower performance levels were less likely to injure themselves per 1000 hours than riders of higher performance levels. Riders who often wore a helmet suffered significantly fewer head injuries and had a significantly lower total injury duration than riders who did not wear a helmet (p = 0.006). Similarly, the study showed that riders who often or always wore a safety vest suffered significantly fewer spinal injuries (p = 0.017) and had significantly fewer injuries per 1000 riding hours (p = 0.031) than riders who did not wear a safety vest.

The main concept of the study is interesting however, there are some minor issues that should be addressed to bring the paper up to publishable standards:

TITLE

I feel that the term ‘overuse damage’ could be changed to ‘overuse injuries’. If the authors do not want to repeat the word ‘injuries’ in the title, please consider rephrasing it.

ABSTRACT

  1. Please describe the methods used to draw the results of the study.

INTRODUCTION

  1. In general: Introduction section is too short. I feel that the authors did not thoroughly read the material used to reference their Introduction. Please, refer to the relevant literature and provide a better rationale for the study.

METHODS

  1. Please provide, the name of the questionnaire used in the study and the relevant reference of the initial publication of the questionnaire. Please, also report the reliability and validity values of the inventory.
  2. How did the authors control for possible bias in the participants answers?
  3. Please, report which parameters were normally distributed, and which parameters were non normally distributed so as the reader can understand the results.
  4. Please, clarify the term ‘unconnected’
  5. How did the authors control for the severity of the injury? Did the authors collect information about recovery and treatment of the injuries?

RESULTS

  1. In general well-written and concise.

DISCUSSION

  1. In general: the attempt at describing the results, is good

Reviewer 3 Report

Thank you for your interesting and generally well written paper for review. I only have a very few questions and comments.

Title

I know this is a human journal, but for clarity to maximise searching, is it worth making it clear that this is a study of the human athlete within show jumping?

Abstract

I think it would be useful to make it clear what the 1000 hours consisted of. Is this competing and training? Many top-level riders would not jump their top horses in training for example, only in competition and in all affiliated SJ events under FEI rules helmets are compulsory, so those not wearing helmets must be during training if professional. This clarity would really help place your work in context within the abstract.

Introduction

You talk about helmet requirements and about rider levels, but these are really specific to one country rather than internationally, so either make this clear in the title i.e. which country you were researching in, or make it clear in the introduction that these rules are in country and which country this is.

I think for the benefit of the readers of this journal who may be completely unfamiliar with equestrian sports, a more in depth and detailed review of the associated literature would be useful and a more detailed description of what the sport actually entails e.g. fence heights, training at home requirements etc. This would build a better picture of what these riders are actually doing within their sporting and training endeavours.

I think it would also be worth referring to the fact that competitive equestrians can have a much longer competitive career than other athletes as this will help place your overuse injury data in context.

Materials and methods

Where were these riders recruited to fill in the survey? You mention that the completion was overseen by medical staff so had the rider been injured or did you go to shows and yards to access these riders? Your recruitment strategy is really important.

You talk about the skill levels, but these are not common across all countries, so I think you need a citation here to allow the reader to follow this up and find out what these skill levels mean, or you need to include this information.

When you say “every 1000 jumping hours” is this training as well as many top riders do not jump their top horses at home, so are these really jumping hours or are they riding hours? You should also specify how you catergorised your riders into the LK groups, I am assuming you did this via their S category, but this needs to be clearer as there seems to be 7 of these and only 6 LK categories.

Results

Was your average exposure time during their entire career?

Is LK1 the highest performance level and LK6 the lowest, I’m afraid this is not made clear in the method or the results.

3.1.1 You state men have more head injuries than women, is this per 1000 hours of riding? I think this could be restated as you said just before that point that men rode significantly more than women. You also say I this section that when considering 1000 hours no significant difference is shown by LK level, I think you need to explain this more clearly. This is clearer on your graph, so I think it needs to ne explained more clearly in the text.

3.4 when analysing the data on safety vest wear, did you consider the rider level as safety vest wearing in showjumping does tend to be associated with the lower level of rider?

Discussion

Do you think that not differentiating between amateur and professional riders may have skewed your data as professional riders will be training and bringing on multiple young horses (possibly) where amateurs will be riding less horses and may not have exposure to the same level of young horse training? Or alternatively, those riders at the very top of the sport may have home riders, so do not need to train their own young horses, so may have less exposure to the dangers of backing young horses?

4.1 Sorry if I have missed it in the results, but you state “The lower injury rate in show jumping by more professional riders was confirmed in the present study” but I could not find this reported in your results. You also state “Injuries in equestrian sports must be considered as particularly severe injuries compared to other sports and are comparable to injuries in skating or cycling” but in the results, you did not really report on what you categorised as severe, would this be worth including?

Overall, I think the discussion needs to have greater clarity throughout to give the reader some clear take home messages.
